# Implementation and Performance Evaluation of Integrated Wireless MultiSensor Module for Aseptic Incubator of *Cordyceps militaris*

**DOI:** 10.3390/s20154272

**Published:** 2020-07-31

**Authors:** Jen-Yung Lin, Huan-Liang Tsai, Wen-Chi Sang

**Affiliations:** 1Department of Computer Science and Information Engineering, Da-Yeh University, Changhua 51591, Taiwan; jylin@mail.dyu.edu.tw (J.-Y.L.); R0806003@cloud.dyu.edu.tw (W.-C.S.); 2School of Engineering, Da-Yeh University, Changhua 51591, Taiwan

**Keywords:** wireless multisensor system, sterilized jar incubator, solid-state fermentation (SSF), *Cordyceps militaris* culture

## Abstract

This paper originally proposes a wireless multisensor module with illuminance, temperature, relative humidity (RH) and carbon dioxide (CO_2_) sensors in an aseptic jar incubator for a solid-state fermentation (SSF) of *Cordyceps militaris* culture. The light intensity, ambient temperature, RH and CO_2_ are the critical cultivation factors of *C. militaris*. First, these sensors are integrated in a multisensor platform which is installed inside a lid and covered with a high-efficiency particulate air (HEPA) membrane of class H14 for sterilization of bacteria and viruses. The observations of sensors are then transmitted by a wireless XBee network where the slave sensor node is fixed at the top of jar lid and the master radio node receives data and uploads to an on-site monitoring node. The acquired information is further transmitted to an iCloud database and displayed in a web-based monitoring system. The results illustrate the proposed wireless multisensor module was validated with sufficient accuracy, reliable confidence and well-tolerance for *C. militaris* cultivation biotechnology under aseptic conditions.

## 1. Introduction

Nowadays, edible and medicinal mushrooms have been recognized as valuable and healthy sources with various biometabolites and bioactive ingredients for foods and possible treatment for diseases. In particular, *Cordyceps militaris* is one of the most important species of *Cordyceps* genus that is a well-known traditional Chinese medicinal mushroom with therapeutically bioactive metabolites such as cordycepin, adenosine, cordymin and exopolysaccharides (EPS). *Cordyceps militaris* has been recognized as an alternative of *Cordyceps sinensis* and has attracted research interests in therapeutic applications with the same sources of phytochemical and biologic constituents [1]. Furthermore, with the advantages of its high nutritional and therapeutic value, *C. militaris* can be directly used as an edible mushroom for food tonics and health food. With increasing standards of living, there is a promising requirement in both quantity and quality for *C. militaris*. However, faced with changes of the natural environment in its host and living conditions, the natural resources of *C. militaris* cannot meet the ever-increasing demand. Therefore, artificial cultivation of *C. militaris* is of increasing importance.

Recently, *C. militaris* cultivation has been extensively studied by solid and liquid fermentation microbial methods [1]. Liquid-fermentation culture features shorter time and higher mycelial production. The highest cordycepin production with the record of 8.57 g/L has been demonstrated by Das et al. [2] using *C. militaris* mutant in a surface liquid culture (SLC) after adding adenosine precursor. Furthermore, Kang et al. [3] demonstrated an optimal condition for the batch culture using SLC technique in a 1000-mL glass jar. The maximum production of cordycepin is 2008.48 mg/L with the most efficient working volume of 700 mL In general, it takes several days to obtain a fruiting body by solid-state fermentation (SSF), and production is not cost-effective. However, a repeated large-scale batch culture featuring the greatest gain in productivity has overcome the bottleneck of SSF method for *C. militaris* culture. Furthermore, nature substrate in SSF process has more biological advantages compared to some chemically defined or biochemical substrate supplements used in commercial *C. militaris* cultivation. Xie et al. [4] studied an optimal ratio of nature materials such as brown rice, malt and soy bean to cultivate in submerge fermentation for *C. militaris*. Wen et al. [5] investigates the optimization of *C. militaris* culture on natural substrate using SSF in batch for fruiting body growth and cordycepin production. The results revealed an increase of fruiting body yield to 67.96% (1.73 ± 0.08 g/bottle) and cordycepin content in fruiting body to 63.17% (9.17 ± 0.09 mg/g). Gregori [6] revealed the cordycepin of 781.11 mg/g with the CM11 strain fungal biomass in the 50% spent brewery grains using SSF process. Furthermore, Adnan et al. [7] demonstrated the SSF using solid substrates of rice, oat and wheat and had cordycepin production of 814.60 mg/g, 638.85 mg/g, and 565.20 mg/g, respectively. Therefore, large-scale industrial cultivation for *C. militaris* using SSF technology with natural substrate is economically viable.

The SSF culture of *C. militaris* is a time-consumption and energy-intensive cultivation process spanning weeks or even months. Considering the long cultivation period of SSF process, the culture automation is of increasing importance. Based on a thorough review process for the recent references [7,8,9,10], an available multisensor device and/or system was not proposed yet so far to our best knowledge. For the SSF culture of *C. militaris*, illuminance, ambient temperature, relative humidity (RH) and carbon dioxide (CO_2_) are the key factors of fruiting body growth. In order to promote both production efficiency and product quality for *C. militaris* cultivation, developing an on-site monitoring system would cost-effectively solve the problems of labor, storage, issues of quantity and quality and energy consumption the same as ones in the agriculture field. Individual sensors and instruments to measure only a single parameter each for all of the above-mentioned parameters have being commercially available. The integration of multiple sensors into a compact multisensor module could scale down both setup complexity and cost and make it feasible to introduce the sensors into a cultivation jar. Meanwhile, an integrated multisensor module is the fundamental sensor to develop a wireless sensor network (WSN) platform for an industrial scale of *C. militaris* cultivation, especially in-field wireless sensors for sterilized incubator like jars or bottles. The main contribution of this paper is aimed to address the design, implementation and evaluation of an on-site wireless multisensor module for the sterilized incubator of *C. militaris* culture. The light intensity, temperature, RH and CO_2_ sensors are integrated with a pair of XBee module for a wireless multisensor module for the jar incubator of *C. militaris* culture. Bothe accuracy and reliance of the proposed module will be validated through the pre-calibration, on-site monitoring and post-calibration demonstrations.

## 2. Novel Wireless Multisensor Module Design

The SSF culture process of *C. militaris* includes initially cultivating mycelium in the incubator jar, following the dark and light cultivation of mycelium, light breeding of primordial, light treatment of fruiting body and harvest of fruiting body. The whole course of SSF culture process of *C. militaris* was conducted in the growth chamber GC-101H [11]. The cultivation conditions for the whole course of SSF culture process of *C. militaris* are listed in Table 1. In order to achieve an industrial scale of automatic *C. militaris* cultivation, a wireless sensor network (WSN) system with multisensor modules for the on-site monitoring of culture environment is developed as shown in Figure 1. This work is aimed to focus on the wireless multisensor modules with illuminance, temperature, RH and CO_2_ sensors for the sensor layer of the WSN system for on-site monitoring the *C. militaris* culture.

### 2.1. Hardware Design for Novel Wireless Multisensor Module

The innovation patent of the novel integrated multisensor module for sterilized incubators was issued on 11 May 2019 in Taiwan [12]. As shown in Figure 2a, the illuminance, temperature, RH and CO_2_ sensors are integrated in the multisensor module. Taking simplicity and cost-effectiveness into consideration, a lower-speed inter-integrated circuit (I^2^C) bus controlled by a commercially available micro-controller unit (MCU) was adopted for a multi-slave, single-ended and series package-switched data transmission. In addition, a multi-time programming memory was used as buffering through an advanced peripheral bus (APB) interconnection. A digital ambient light sensor “BH1750FVI” [13] with an I^2^C bus interface was adopted to measure the illuminance with wide range of 1-65535 lux and high resolution of 1 lux. A temperature/RH sensor “ENS210” [14] was used to acquire the ambient temperature and humidity. The operation temperature ranges from −40 °C to 100 °C with the resolution of 0.016 °C and the accuracy of ±0.2 °C (@25 °C) The RH operation range is between 0–100% with the resolution of 0.03% and the accuracy of ±5% (@25 °C, 0–100%). A commercially available non-dispersive infrared (NDIR) CO_2_ sensor with the measurement range of 400–10,000 ppm and the accuracy of ±30 ppm was integrated for CO_2_ detection, which was built in the SCD30 sensor module [15]. All of the four sensors have the same I^2^C digital interface. In order to connect the following Arduino-based prototype system for wireless communication, the wiring scheme of “Vcc, SDA (series data line), GND, SCL (series clock line)” was arranged in the form of alternating signal and power connections. The sensor layout arrangement and external wireless configuration of the proposed wireless multisensor module are shown in Figure 2c. As depicted in in Figure 3, a tie-in hinged lid of the jar was custom-designed. In order to connect the internal multisensor module with the external Arduino-bases wireless system, a successive wiring connection of “V_DD_ (as “Vcc”), GND, SCL, SDA” in series was well-defined and the corresponding holes in the jar cap were designed for installation. An Arduino-based system with an ATMega328 MCU was developed and a low-power ZigBee module (XBee) as a slave node was integrated for wireless communication. As shown in Figure 3, the integrated multisensor module is installed inside jar lid by a 6p-pin header with four pins for I^2^C signal connection and two pins for firmness enhancement. The rear and front viewpoint photographs of the implemented prototype are shown in Figure 3d.

All equipment introduced in the aseptic cultivation of *C. militaris* needs to undergo sterilization process. The sterilization process can be generally classified into physical and chemical sterilizations. Thermal and radiation sterilization are the kind of physical processes. Dry heat and steam heat are common methods of thermal sterilization. Radiation with high energy such as gamma irradiation or electron beam could impair the sensors. Posed to issues of food safety, chemical sterilization is not considered, and the jar incubator of glass is typically sterilized by steam heat (autoclave). The integrated multisensor module with jar incubator were thermally sterilized. The finished lid was then covered with a high-efficiency particulate air (HEPA) membrane of class H14 for sterilization of bacteria and viruses. The HEPA membrane of 70-mm diameter was placed in the bottom ring of the lid as shown in Figure 3 and sealed between glass jar and lid.

On the other hand, the master XBee node as a receiver was connected to local node (laptop computer in the paper) using a USB series communication interface. The hardware implementation of the proposed WSN system is illustrated in Figure 4.

### 2.2. Software Design for Wireless Multisensor System

The software for the proposed wireless multisensor system was developed under the open-source Arduino integrated development environment (IDE) which support the languages C and C++ with specified rules of code structure. The flowchart of data acquisition for the proposed wireless multisensor module is depicted in Figure 5.

## 3. Results and Discussion

The cultivation substrate consists of brown rice with the weight of 25 g and 0.084 g nutrition liquid of 40 mL which were placed in a 300-mL glass bottle. In order to prevent the possible contamination for the SSF culture of *C. militaris*, the cultivation jars, medium and tools being used are sterile to begin with to make the practices succeed. All materials and the proposed multisensor modules in the lids are sterilized by an autoclave that heats the material to high temperature of 120 °C for a time of 20 min long enough to kill all possible microorganisms.

### 3.1. Pre-Calibration of Multisensor Module

The proposed multisensor module was first installed into the lid of incubator jar. Both grass jar and lid were autoclaved for a 20 min at 120 °C thermal sterilization process. In order to double-check the performance of the multisensor module, a calibration measurement was first performed using commercially available measurement instrument. A light meter “LX-1102” [16] (Lutron Electronic, 2015) with the detector area of 1 cm^2^ and accuracy of ±530 lux (@0~50 °C) was used to measure the ambient illuminance. Two Omniport 30 instruments with RH/T and CO_2_ probes [17] (Lutron Elektronics, 2015) were used to calibrate the temperature, RH and CO_2_ sensors, respectively. One Omniport 30 instrument with a Logprobe 16 was used to acquire temperature and RH at the accuracy of ≤±0.5 °C and ≤±3% (@−20~70 °C), respectively. The other one with a Logprobe 810 was used to acquire CO_2_ reading with the accuracy of ±5 ppm (@25 °C, 1013 mbar). The pre-calibration process was conducted for a 5-day continuous measurement in the growth chamber GC-101H. The indoor conditions were set at the illuminance of 1100 lux the temperature of 25 °C the RH of 75% and the CO_2_ concentration of 455 ppm. The observation of all sensors and instruments at a 5-min sampling period for 24-h measurement is depicted in Figure 5. Taking the results of measurement instrument as references, the differences between multisensor module and measurement instrument are also depicted in Figure 6.

Taking the results of measurement instrument as references, the relative difference is defined as:(1)$$e_{i}^{j}=x_{i}^{j}-{\bar{x}}_{i}^{j}$$
where $x_{i}^{j}$ and ${\bar{x}}_{i}^{j}$ are the *j*th measurement values of multisensor module and measurement instruments, $i=\lambda \;(\mathrm{illuminance});\;T\;(\mathrm{temperature});\;\phi \;(\mathrm{RH});\;\eta_{{\mathrm{CO}}_{2}}{\;(\mathrm{CO}}_{2}).$ The differences of illuminance reading between multisensor module and light meter are in the range of −1.8998~1.4457 lux. The differences of temperature and RH between them are −0.9449~0.8069 °C and −1.0404~0.9166%, respectively. The relative differences of CO_2_ are in the range of −1.4414~1.0969 ppm. The difference analysis results between the proposed multisensor module and measurement instruments in the pre-calibration process are listed in Table 2. The results show that these relative differences are all acceptable than the accuracy of measurement instruments. This illustrates the reading accuracy and measurement confidence of the proposed multisensor module.

### 3.2. On-Site Monitoring Results

The whole course of SSF culture of *C. militaris* is shown above. After thermal sterilization, the seed culture of *C. militaris* was incubated on both cultivation substrate of brown rice with the weight of 25 g and 0.084 g nutrition liquid of 40 mL inside a 300-mL jar covered by a lid built into with the multisensor module. The whole-course culture of *C. militaris* was performed in the growth chamber GC-101H. The proposed multisensor module was built in the lid to monitor the inside culture conditions. All on-site observations of illuminance, temperature, RH and CO_2_ were acquired at a 5-min interval and then sent to a local laptop computer by the wireless XBee communication. First, the mycelium was incubated at the rice substrate in a dark treatment at 20 °C from the first day to day 7. Moreover, then a light treatment with illuminance of 500 lux was conducted from day 8 to day 10. Meanwhile, the daytime temperature was maintained at 24 °C with the RH of 70% and the nighttime temperature was lowered to 18 °C with 90% RH. The indoor air was circulated to maintain the CO_2_ level less than 460 ppm. Primordia formation of fruiting body began with day/light duty cycle of 16 h/8 h at 11–20 days. The temperature and RH conditions for the day-/nighttime duty cycle were maintained at 24/18 °C with the RH of 75%/90%, respectively. The illuminance of indoor conditions were set at 1500/0 lux and the air was circulated to maintain the CO_2_ level of 460 ppm. From 21 to 50 days, fruiting body was enhanced to cultivate with the illuminance of 1500/0 lux at day/night duty cycle of 12/12 h. The day-/night–time temperature was controlled at 22/16 °C with the RH of 90%/90%. After the fruiting body culture, the day-/night–time light intensity was maintained at 1000/0 lux at day/night duty cycle of 12/12 h, as well as both temperature and RH were controlled at the same levels for the final harvest stage. A 24-h readings during the primordia culture are depicted in Figure 7. Figure 7a shows that almost no time delay existed in the illuminance exchange for the light intensity of 1500/0 lux at day/night duty cycle of 12/12 h. As shown in Figure 7, a near 10-min delay could be found in the temperature change of 24/18 °C for the day/night duty cycle, which is caused by the heat transfer mechanism. Caused by convention and diffusion mechanisms, the same situation exists in the RH variation as shown in Figure 7. As illustrated in Figure 7, the CO_2_ concentration was maintained less than 460 ppm level with sufficient air exchange by circulation. From the CO_2_ measurements it can be clearly seen raise of CO_2_ levels due to elevated temperature at 08:05 with the light intensity of 1500/0 lux at day/night duty cycle of 12/12 h. The evidence demonstrates the SSF culture of *C. militaris* consumed oxygen and generated CO_2_ Furthermore, a 24-h readings during the fruiting body culture are depicted in Figure 8. As shown in Figure 8, it is obvious that almost no time delay existed in the illuminance exchange for the light intensity of 1500/0 lux at day/light duty cycle of 12/12 h. Due to heat transfer mechanism, there was near 10-min delay in the temperature change of 22/16 °C for the day/night duty cycle as shown in Figure 8. Figure 8c shows the same situation existing in the RH variation, which was caused by convention and diffusion mechanisms. As illustrated in Figure 8, the CO_2_ level was maintained in the range of 5% concentration with sufficient air exchange by circulation. Only the 467.5 ppm reading was acquired at 08:05 with the light intensity of 1500/0 lux at day/light duty cycle of 12/12 h. As compared to Figure 7, both oxygen consumption and CO_2_ generation for the fruiting body culture were greater than ones of primordia culture.

### 3.3. Post-Calibration Results

After near two-month on-site monitoring process, the proposed multisensor module was calibrated with the same light meter, temperature/RH and CO_2_ measurement instruments as illustrated in the pre-calibration process. In order to verify the reliability and confidence of the proposed multisensor module, the post-calibration process was carried out at the environmental conditions with illuminance of 1100 lux, temperature of 25 °C, RH of 75% and CO_2_ concentration of 460 ppm. The measurement results for both multisensor module and measurement instruments are depicted in Figure 9. As compared to these in Figure 7, there are no obvious differences existing for all observation of difference for illuminance, temperature, RH and CO_2_ readings. The detail of sensing differences are also listed in Table 2. Both temperature and RH readings have an increasing variation in difference and the illuminance and CO_2_ difference have a convergence trend. Through both pre- and post-calibration processes under the same environment and conditions, both reliability and confidence were confirmed and the durability of the proposed module was tested through on-site monitoring test over near two months.

### 3.4. Discussion

In order to easily assess the accuracy of the proposed multisensor module, the difference between the measurement results of the proposed module and commercial measurement instruments in pre-calibration and post-calibration processes are shown in Figure 6 and Figure 9. Based on the difference results by taking measurement instruments as references, the corresponding root mean square error (RMSE) are defined as
(2)$$\mathrm{RMSE}(e_{i}^{j})=\sqrt{\sum_{i=1}^{n}{\left(x_{i}^{j}-{\bar{x}}_{i}^{j}\right)}^{2}/n}$$
where n is the total number of observations. As shown in Table 2, the difference analysis between pre- and post-calibration processes reveals no obvious fluctuation existing for illuminance, temperature, RH and CO_2_ readings. Furthermore, Table 2 illustrates no apparent difference in the RMSE analysis. These results further demonstrate the proposed multisensor provides sufficient reliability and durability for the long-term on-site monitoring ability through the comparison between pre- and post-calibration even over months.

So far, the integration of multiple sensors, the XBee wireless communication, an autoclave sterilization of the proposed wireless multisensor module, a durability evaluation for long-term operation with high humidity and an accuracy validation for the on-site measurement were well illustrated in this work. It is obviously found that sensors for different sensing can be integrated in a single module. With the consideration of accuracy, a commercially available NDIR CO_2_ sensor was adopted in the proposed multisensor module. Posed to large volume, a metal oxide chemiresistor with a heater on Si-based MEMS structure will be used to replace the NDIR one for miniature. Through the whole course of overall *C. militaris* cultivation, light, oxygen, temperature and water are the main controllable factors. As shown in Figure 7 and Figure 8d, it is obviously found that a peak increase of CO_2_ in response to day–time cultivation reveals *C. militaris* culture released CO_2_ with O_2_ consumption and the CO_2_ concentration for the fruiting body culture are greater than ones of primordia culture. The issue of CO_2_-level control during the long-term fruiting body cultivation could be evaluated in the future.

Even having sufficient performance in the electrical characteristics, the possible contamination in the SSF culture process of *C. militaris* is possibly caused by the on-site multisensor module. The class H14 polyurethane HEPA membrane in the lid was double-checked by visual inspection. No obvious spot of contamination was found in the HEPA membrane surface. The possible pollution by the leakage and corrosion of substance from the multisensor module was also verified before the post-calibration. The results found no possible pollution is caused by the proposed multisensor module after long-term sterile culture of *C. militaris*. The aging issue of the proposed multisensor module could be further verified.

## 4. Conclusions

A wireless multisensor system integrates the all illuminance, temperature, relative humidity (RH) and carbon dioxide (CO_2_) sensors with XBee wireless communication platform for an aseptic jar incubator was illustrated and evaluated with sufficient accuracy and confidence. The proposed multisensor module has the well-tolerance performance of sterile cultivation for a solid-state fermentation (SSF) of *Cordyceps militaris*. From the viewpoint of industrial mass-production for a SSF process of *C. militaris*, the proposed wireless multisensor module features: (1) much simpler setup and reuse, more cost-effectiveness than multiple single sensors without external sample-taking instrument and wiring; (2) sufficient accuracy and reliability with pre-calibration even for commercialized sensor devices; (3) simultaneously on-site monitoring of multiple sensing parameters in close proximity to, but without direct contact with *C. militaris* over weeks or even months. Thanks to the promising development of precision agriculture, the proposed platform will provide a flexible and expandable biotechnology development to simultaneously monitor a wide range of cultivation parameters in agricultural culture with transparency and quality control through the whole process. The issue of sensor aging will be further considered for measurement robustness.

## Figures and Tables

**Figure 1 sensors-20-04272-f001:**
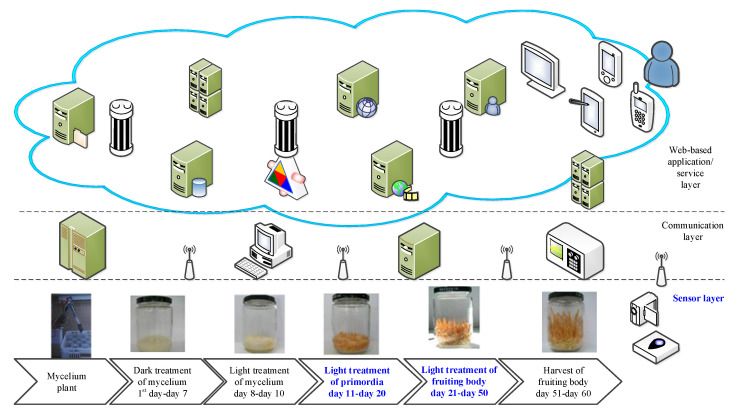
Configuration of wireless sensor network (WSN) system for *C. militaris* culture.

**Figure 2 sensors-20-04272-f002:**
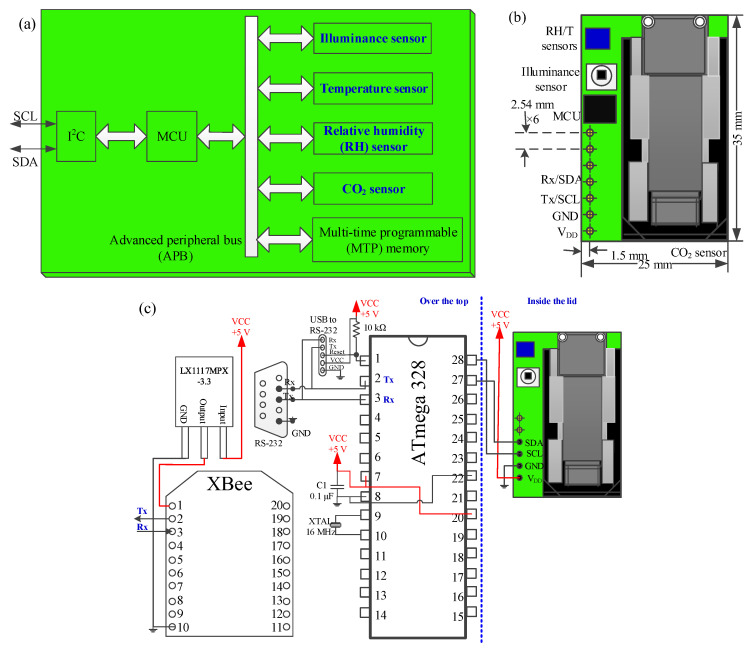
Schematic of wireless multisensor module. (**a**) Function block; (**b**) sensor layout arrangement; (**c**) wireless configuration.

**Figure 3 sensors-20-04272-f003:**
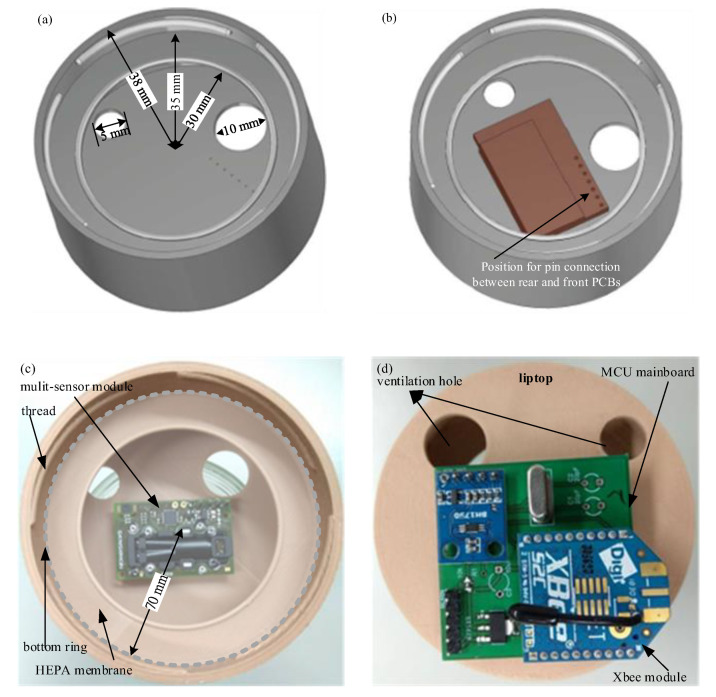
Schematic of jar lid. (**a**) Layout; (**b**) multisensor module arrangement; (**c**) rear-view and (**d**) front-view prototype photographs of hardware implementation for proposed multisensor module.

**Figure 4 sensors-20-04272-f004:**
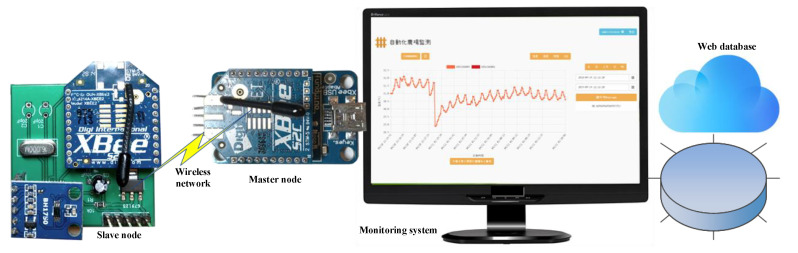
Hardware implementation of proposed WSN system.

**Figure 5 sensors-20-04272-f005:**
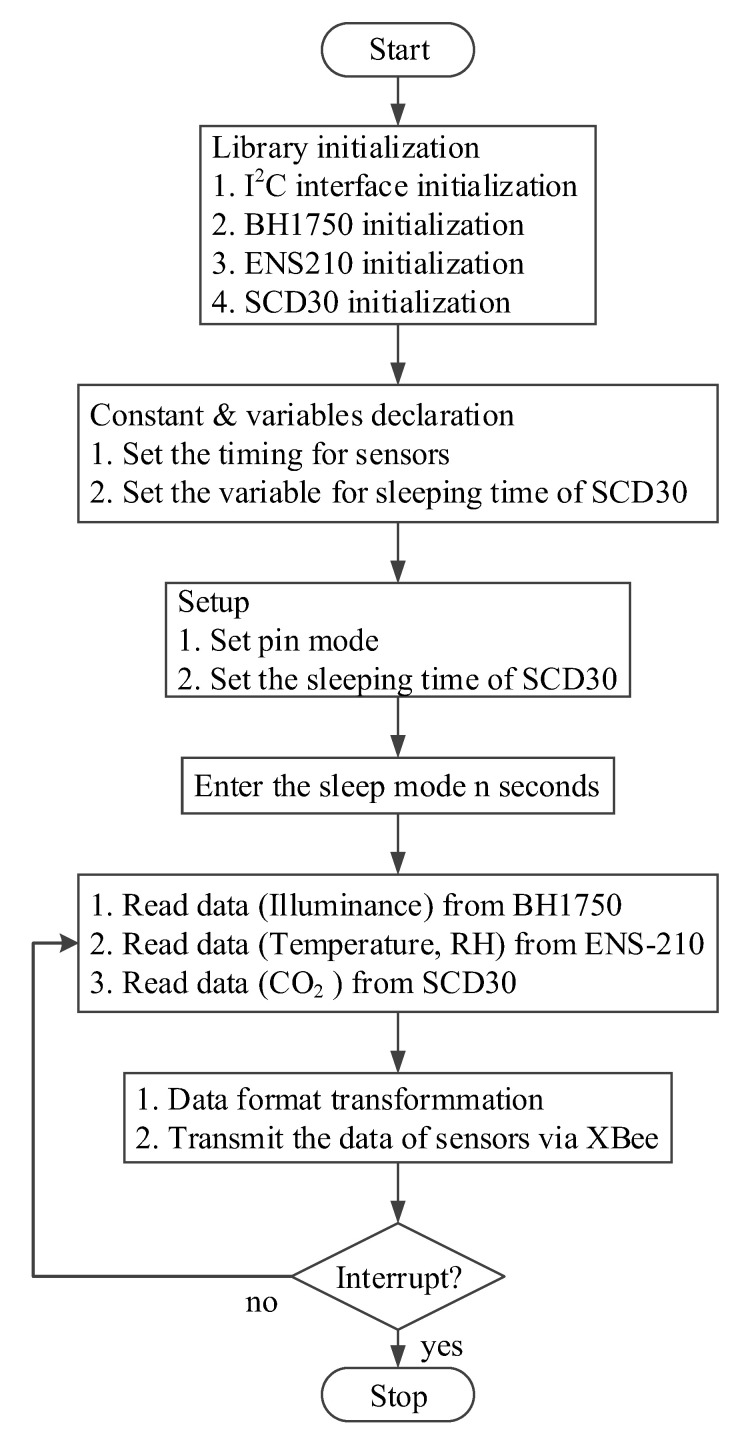
Flowchart of data acquisition for proposed wireless multisensor module.

**Figure 6 sensors-20-04272-f006:**
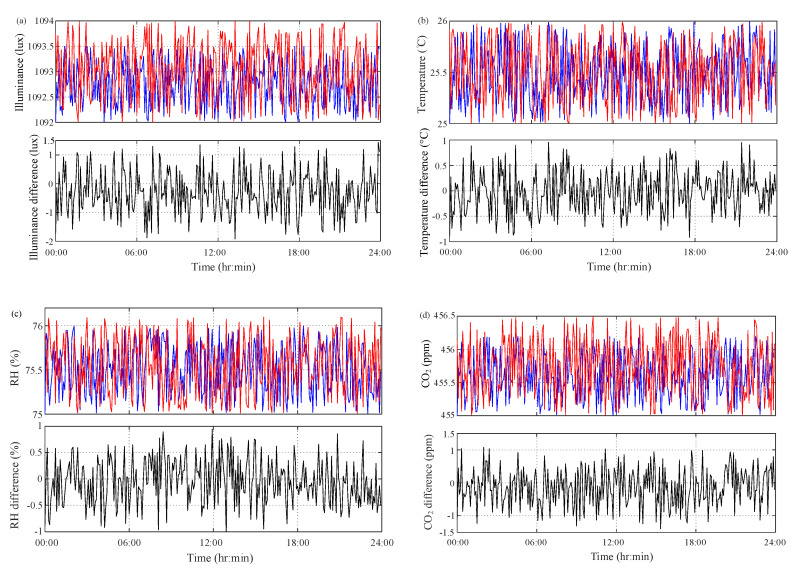
24-h measurement results of pre-calibration process. (**a**) Illuminance; (**b**) temperature; (**c**) relative humidity (RH); (**d**) CO_2_ (blue line—sensor; red line—instrument; black line—difference).

**Figure 7 sensors-20-04272-f007:**
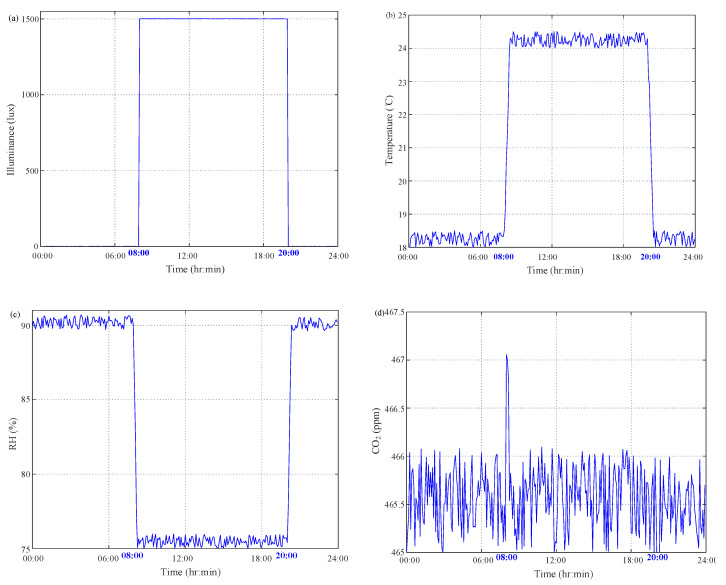
24-h measurement results of on-site monitoring for primordia culture. (**a**) Illuminance; (**b**) temperature; (**c**) RH; (**d**) CO_2_.

**Figure 8 sensors-20-04272-f008:**
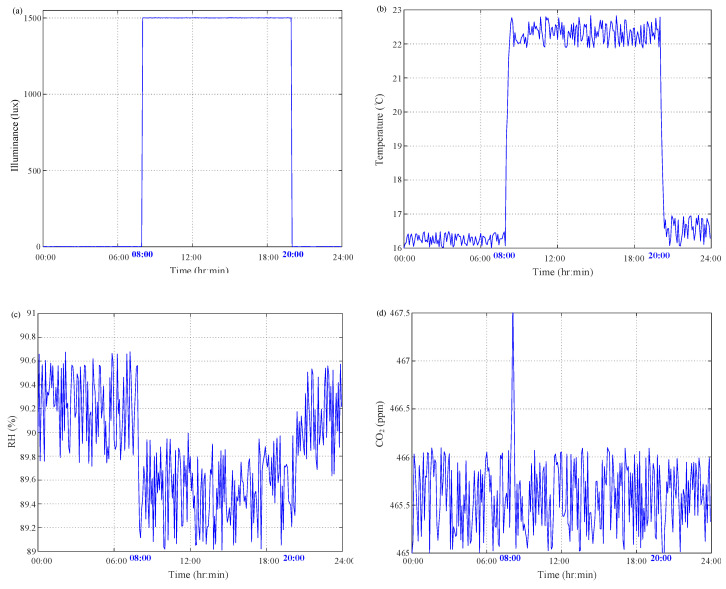
24-h measurement results of on-site monitoring for fruiting body culture. (**a**) Illuminance; (**b**) temperature; (**c**) RH; (**d**) CO_2_.

**Figure 9 sensors-20-04272-f009:**
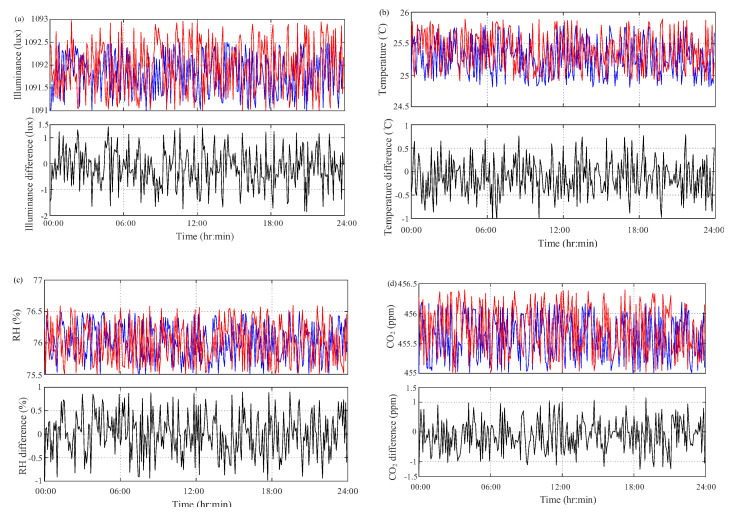
24-h measurement results of post-calibration process: (**a**) illuminance; (**b**) temperature; (**c**) RH; (**d**) CO_2_ (blue line—sensor; red line—instrument; black line—difference).

**Table 1 sensors-20-04272-t001:** Cultivation conditions for whole course of solid-state fermentation (SSF) culture process of *C. militaris*.

Culture Stage	Duration	Day/Night Duty Cycle	Illuminance (lux)	Temperature (°C)	RH (%)	CO_2_ (ppm)
Dark culutre of mycelium	1st day–day 7	0/24	0	20	60	airtight
Light culutre of mycelium	day 8–day 10	8/16	1000/500	24/18	70/90	460
Primordia culture	day 11–day 20	12/12	1000/500	24/18	75/90	460
Fruiting-body culture	day 21–day 50	12/12	1000/500	22/16	90/90	460
Fruiting-body harvest	day 51–day 60	12/12	1000/500	22/16	90/90	460

**Table 2 sensors-20-04272-t002:** Accuracy analysis of the proposed multisensor module.

Items		Pre-Calibration	Post-Calibration
Relative erros	*e*(∆*λ*) (lux)	−1.8998~1.4457	−1.8513~1.4224
	*e*(∆*T*) (°C)	−0.9449~0..8069	−1.8513~1.4224
	*e*(∆ψ) (%)	−1.0404~0.9166	−1.3867~0.8874
	*e*(∆η_CO2_) (ppm)	−1.4414~1.0969	−1.3697~1.0886
Root mean square error (RMSE)	RMSE(∆*λ*) (lux)	0.7802	0.7984
	RMSE (∆*T*) (°C)	0.3970	0.4047
	RMSE (∆ψ) (%)	0.4673	0.4468
	RMSE (∆η_CO2_) (ppm)	0.5838	0.5035

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
