# Peer review of "Implementation and Performance Evaluation of Integrated Wireless MultiSensor Module for Aseptic Incubator of Cordyceps militaris"

_sensors, 2020, doi:10.3390/s20154272_

Round 1
Reviewer 1 Report
A very interesting and useful presentation of simple measurement of growth conditions in the fungus growth vessel. A kind of detection all mushroom growers were waiting for.
As all tests were conducted in a satisfying manner according to my knowledge. I am a bit concerned about the accuracy of measured CO2 levels as they seem very low. In our 1L experimental cultivation vessels equiped with 15mm diameter HEPA class 13 membrane, containing 400g of substrate inoculated with c. militaris CO2 levels raise up to 20.000 ppm compared to 467,5ppm in this manuscript. This seems as a too high difference to ignore. To solve this question, please add into the manuscript details as are diameter of the HEPA filters, weight of cultivation substrate, c. militaris strain used, etc., which could answer the question of very low CO2 production by mycelia or its difusion into surrounding. I am speculating that sensor is placed too close to the filters and is thus measuring ambiental CO2 levels instead of levels inside the vessel, or filters are too large. This is one of the main issues that need to be cleared out. One of possibilities is to measure CO2 levels on different heights inside the vessel and compare them with measurements of a presented multi-sensor module.
These are the details that need to be corrected as well:
52,59-natural
60,61-...is a time and energy consuming cultivation process.
74-in field?
86-in table
88-in field
103, 104-delete RH
123-such as gamma
126-It is not clear where do HEPA filters fit onto the lid. Please explain in text as well as indicate in Figure 3.
145-20 minutes at 120
147-performed
150-Lutron Elektronics
152-at the accuracy
165-differences of CO2
166-delete "The root mean square error"
173-delete "natural substrate" and specify the mass of wet substrate, its water content
178-This sentence needs to be corrected as it is not clear
180-from day 8 to day 10 - correct all in the same manner
182, 205-5% CO2 is 5.000 ppm but you specify it was maintained at 460ppm
197-Please describe as follows - From the CO2 measurements it can be clearly seen raise of CO2 levels due to elevated temperature at 08.05 hours.
214-what kind of environment and conditions?
221-was tested
FIG2-CO2 sensor is not depicted?
FIG3-the letters on pictures are missing
-why is lid on first picture higher than the one on second?
-please specify the location of the HEPA membranes
-specify which is the upper and which is the bottom part of the lid
-specify the diameter of the membranes and its technical characteristics
-depict how is the lower sensor connected with the upper PCB
TABLE1-please add on second place a column with days from the day of inoculation
-move columns so that "illuminance" and "day/night" are located together
Author Response
Dear Reviewer#1:
I sincerely appreciate your valuable comments and suggestions. The response is attached. All of the correction and modification are highlighted in red in the revised manuscript. Thank you a million.
Best Regards
Huan-Liang Tsai
Professor
School of Engineering,
Da-Yeh University, Taiwan

Reviewer 2 Report
Overall comments:
This paper deals with a wireless multi-sensor module for the maintenance of optimum C. militaris culture condition.
It is considered that the paper is well written and would be qualified to ensure its publication after proper revision of following minor concerns.
Minor concerns:
- Fig 7(c) and Fig 8(c) have different y-axis scale. Thus, Fig 8(c) seems to be more fluctuated. Is there any reason for that?
- We are curious about how much the population of C. militaris was maintained or increased in the presented culture system compared to the conventional culture system. It is recommended that the authors provide, if any, additional experiments to quantify and/or validate it.
Author Response
Dear Reviewer#2:
I sincerely appreciate your valuable comments and suggestions. The response is attached. All of the correction and modification are highlighted in red in the revised manuscript. Thank you a million.
Best Regards
Huan-Liang Tsai
Professor
School of Engineering,
Da-Yeh University, Taiwan

Reviewer 3 Report
The manuscript “Implementation and Performance Evaluation of Integrated Wireless Multi-Sensor Module for Aseptic Incubator of Cordyceps militaris” submitted to sensors contributes to our knowledge of wireless sensing system for cost-effective production of Cordyceps militaris fruiting bodies. However, some comments are made below to improve the manuscript.
English improvement is highly recommended.
Lines 12, 23, 28 and 248, “Cordyceps militaris (C. militaris)” is not a good format. Delete “(C. militaris)”.
Line 30, when beginning with sentence, start with “Cordyceps militaris”, not “C. militaris”.
Line 31, “alternative to Cordyceps”. It should be “alternative to Cordyceps sinensis”. Give a reference for this statement.
Lines 55-58, please cross-check the amount of cordycepin production with the authors of the references [6] and [7]. Usually, literature report less than 10 mg/g of cordycepin production by C. militaris.
Lines 182, 196 and 205, CO2 is calculated as 5%. Can the authors change to ppm? In Fig. 7(d), it is given as ppm.
Discussion part is too short. Elaborate some of the points of the research, implications and future prospects.
Lines 258-259, the title of the section and the content do not match with each other.
Line 263, make “Cordyceps militaris” italic.
The impacts of illuminance, temp. and RH on fruiting of C. militaris have been much discussed in the past. Can the authors specifically discuss some impact of CO2 on fruiting of C. militaris?
Lines 332-333, label a, b, c and d in Fig. 3.

Author Response
Dear Reviewer#3:
I sincerely appreciate your valuable comments and suggestions. The response is attached. All of the correction and modification are highlighted in red in the revised manuscript. Thank you a million.
Best Regards
Huan-Liang Tsai
Professor
School of Engineering,
Da-Yeh University, Taiwan
